

# Prediction models of macro-nutrient content in plant organs of *Cucumis melo* in response to soil elements using support vector regression

Abbas Keshtehgar, Mahdi Dahmardeh, Ahmad Ghanbari and Issa Khammari

Department of Agronomy, University of Zabol, Zabol, Sistan and Baluchestan, Iran

Corresponding author
Mahdi Dahmardeh,
dr.dahmardeh@uoz.ac.ir

## ABSTRACT

**Background**. Undoubtedly, the importance of food and food security as one of the present and future challenges is not invisible to anyone. Nowadays, the development of methods for monitoring the nutrient content in crop products is an essential issue for implementing reasonable and logical soil properties management. The modeling technique can evaluate the soil properties of fields and study the subject of crop yield through soil management. This study aims to predict fruit yield and macro-nutrient content in plant organs of *Cucumis melo* in response to soil elements using support vector regression (SVR).

**Methodology**. In the spring of 2020, this study was done as a factorial test in a randomized complete block design with three replications. The first factor was the use of fertilizers in six levels: no fertilizer (control), cow manure (30 t ha$^{-1}$), sheep manure (30 t ha$^{-1}$), nanobiomic foliar application (2 l ha$^{-1}$), silicone foliar application (3 l ha$^{-1}$), and chemical fertilizer from urea, triple superphosphate, and potassium sulfate sources (200, 100, and 150 kg ha$^{-1}$). In addition, four levels of vermicompost considering as the second factor: no vermicompost (control), 5, 10, and 15 t ha$^{-1}$. Input data sets such as fruit yield and nitrogen, phosphorus, and potassium levels in the seeds, fruits, leaves, and roots are used to calibrate the probabilistic model of SP using SVR.

**Results**. According to the results, when the data sets of the nitrogen, phosphorus, and potassium in the fruit uses as input, the accuracy of these models was higher than 80.0% ($R^2 = 0.807$ for predicting fruit nitrogen; $R^2 = 0.999$ for fruit phosphorus; $R^2 = 0.968$ for fruit potassium). Also, the results of the prediction models in response to soil elements showed that the soil nitrogen content ranged from 0.05 to 1.1%, soil phosphorus from 10 to 59 mg kg$^{-1}$, and soil potassium from 180 to 320 mg kg$^{-1}$, which offers a suitable macro-nutrient content in the soil. Likewise, the best fruit nitrogen content ranged from 1.27 to 4.33%, fruit phosphorus from 15.74 to 26.19%, fruit potassium from 15.19 to 19.67%, and fruit yield from 2.16 to 5.95 kg per plant obtained under NPK chemical fertilizers and using 15 t ha$^{-1}$ of vermicompost.

**Conclusions**. Because the fruit values had the highest contribution in prediction than observed values, thus identified as the best plant organs in response to soil elements. Based on our findings, the importance of fruit phosphorus identifies as a determinant that strongly influenced melon prediction models. More significant values of soil elements do not affect increasing fruit yield and macro-nutrient content in plant organs, and excessive application may not be economical. Therefore, our studies provide an

efficient approach with potentially high accuracy to estimate fruit yield and macro-nutrient in the fruits of *Cucumis melo* in response to soil elements and cause a saving in the amount of fertilizer during the growing season.

## INTRODUCTION

Melon (*Cucumis melo* L.), a member of the *Cucurbitaceae* family, is one of the most important vegetable crops worldwide. The major melon producers are China, Turkey, Iran, India, Kazakhstan, and the United States of America (*Food and Agriculture Organization, 2018*). *Cucumis melo* L. ($2n = 2x = 24$) has grown in various geographical areas of Iran from historical eras (*Munger & Robinson, 1991*). Based on archaeological evidence, Iran has been an important center of domestication since 5,000 years ago (*Bisognin, 2002*). Melon is the most polymorphic species of the cucurbit family, which is particularly true for fruit-related traits (*Luan et al., 2010*).

In most melons that belong to the *Cucurbitaceae* family, nutrient requirements and NPK ratio significantly vary depending on the melon type and cultivar, soil mineral status, and crop growth stage (*Deus et al., 2015*; *Chen et al., 2019*). Nitrogen is the most needed macro-nutrient in all cropping systems due to its chief role in the biochemical and physiological processes of the plant (*Pourranjbari Saghaiesh, Souri & Moghaddam, 2019*). Nitrogen is essential during the vegetative phase for the adequate canopy and leaf area to ensure yield capacity. However, excess nitrogen availability during the reproductive phase promotes undesired competition between fruit and vegetation, which might reduce produce quality (*Ferrante et al., 2008*). Likewise, phosphorus is macro-nutrient have different roles in plant metabolism (*Pourranjbari Saghaiesh, Souri & Moghaddam, 2019*). Thus, phosphorus is mainly required for seedling establishment (root growth) and then at early reproductive steps (bloom and seed development) (*Martuscelli et al., 2016*; *Chen et al., 2019*). It is a fact that under unsuitable soil conditions, nitrogen and phosphorus fertilizers can develop plant root growth and biological nitrogen fixation efficiency at low rates (*Pourranjbari Saghaiesh, Souri & Moghaddam, 2019*). Also, potassium is most efficient during the later stages of fruit development, supporting sugar translocation and accumulation (*Deus et al., 2015*; *Tränkner, Tavakol & Jákli, 2018*).

Modeling techniques in farm management can improve performance and economic returns by optimizing crop inputs (fertilizers and chemicals) and preserving the environment and energy resources (water resources, *etc.*). The modeling technique has many benefits, including the capacity to predict numerous soil parameters and perform measurements in labs and farms (*Viscarra Rossel et al., 2006*) and not use chemicals (*Stenberg et al., 2007*). Also, crop product monitoring allows farmers to conduct proper farming operations during the growing season.

Accordingly, data-driven models are needed to efficiently link input data to the desired output (*Adeyemi et al., 2018*). The benefits of support vector machines on artificial neural networks identify in many types of research, which has attracted much research attention (*Jiang et al., 2019*). The structure and performance of support vector machines have been the main target of many studies (*Roodposhti, Safarrad & Shahabi, 2017*).

The study of *Esfandiarpour-Boroujeni et al. (2019)* aimed to evaluate the performance of a hybrid particle swarm optimization-imperialist competitive algorithm-support vector regression (PSO-ICA-SVR) method to predict apricot yield and identify important factors in the Abarkuh, Yazd, Iran. The validation results showed that the hybrid algorithm estimated relatively high accuracy (root mean square error = 1.737 for training data and 2.329 for testing data) for apricot yield. Likewise, *Jeong et al. (2017)* estimated the organic matter, soil available potassium, and phosphorus using support vector machine models. They found that the predicted and actual parameters had a strong correlation.

Prediction of active ingredients in *Salvia miltiorrhiza* Bunge. based on soil elements and artificial neural networks was performed by *Liu et al. (2022)*. This study measured the active ingredients in the roots of *S. miltiorrhiza* and the contents of rhizosphere soil elements from 25 production areas in eight provinces of China and data used for developing a prediction model based on BP (back-propagation) neural network. The results showed that the active ingredients had different degrees of correlation with soil macro-nutrients and trace elements, and the prediction model had the best performance (Mean Square Error = 0.0203, 0.0164; $R^2$ = 0.93, 0.94).

*Mohamed et al. (2021)* performed a field experiment to investigate the use of phosphorus fertilizer source in common bean (*Phaseolus vulgaris* L.) cultivated under salinity stress. The response curve of total dry weight to different rates of phosphorus proved that the quadratic model fit better than the linear model for phosphorus sources. The total dry weight was predicted at 1.675 t ha$^{-1}$ for superphosphate and 1.875 t ha$^{-1}$ for urea phosphate when phosphorus using at 51.5 kg ha$^{-1}$ and 42.5 kg ha$^{-1}$, respectively. In conclusion, the 35.0 kg ha$^{-1}$ phosphorus could be considered the most efficient phosphorus level.

The previous studies recognized partial information on support vector regression models to predict fruit yield and macro-nutrient content in *Cucumis melo* plant organs in response to soil elements. Therefore, the present study aimed to determine: (i) regression models to predict fruit yield and macro-nutrient content in plant organs in response to soil elements; (ii) the effect of soil elements on fruit yield and macro-nutrient content in plant organs; (iii) determinants of prediction fruit yield and macro-nutrient related to different fertilizer used and their levels, and (iv) optimization of the fertilizer used in a cropping system, taking into account the levels of macro-nutrients in the plant organs, fruit yield, and soil elements.

## MATERIALS AND METHODS

### Geographical location and meteorological information of the test site

In the spring of 2020, this study conducting in two Fariman and Zahak counties. Fariman county is situated in northeastern Iran at 35°70′N and 59°85′E, at an altitude

of 1,403 m above sea level, in the hot and dry Mediterranean climates based on the Köppen classification (https://en.db-city.com/Iran--Khorasan-Razavi--Fariman). Zahak County, too, is situated in southeastern Iran at 30°89′N and 61°70′E, at an altitude of 483 m above sea level in the hot and dry climates based on the Köppen classification (https://en.db-city.com/Iran--Sistan-Va-Baluchestan--Zehak).

## Preparation of soil samples

Before beginning the experiment, ten samples were randomly collected from 0 to 30 centimeters in depth to evaluate the chemical properties and soil elements composition. Table S1 presents the results of the soil sample test. To investigate the electrical conductivity and acidity of the soil (as too for the vermicompost and manure used in the study), saturation extract in first made from all the samples, and then the electrical conductivity and acidity were estimated by using the EC meter (Iuchi meter, CyberScan 100, TDS/Conductivity, Singapore), and pH meter (Horiba pH meter, model D-52, Kyoto, Japan) and placed the device electrodes in the saturation extract. To evaluate the nitrogen, phosphorus, and for potassium, samples were placed in Avon Digital (model PTN 55, Pars Teb Novin, Iran) at 70 °C for 48 h, and their dried sample preparing. The nitrogen was measured by using Kjeldahl's (1883) method, phosphorus using Olsen et al. (1954) method by Spectrophotometer (model UV-2100S, Unico, Franksville, WI, USA), and potassium using the Flame Photometer (model PFP7, Jenway, London, England) in the laboratory.

## Experimental design

This study used the support vector regression (SVR) to predict fruit yield and macro-nutrient content in *Cucumis melo* plant organs in response to soil elements affected by different fertilizers as a factorial test in the form of a randomized complete block design with three replications. The first factor was the use of fertilizers in six levels: no fertilizer (control), cow manure (30 t ha$^{-1}$), sheep manure (30 t ha$^{-1}$), nanobiomic foliar application (2 l ha$^{-1}$), silicone foliar application (3 l ha$^{-1}$), and chemical fertilizer from urea, triple superphosphate, and potassium sulfate sources (200, 100, and 150 kg ha$^{-1}$). In addition, four levels of vermicompost considering as the second factor: no vermicompost (control), 5, 10, and 15 t ha$^{-1}$.

## Planting operation

Before planting and in the fall, 30 t ha$^{-1}$ cow manure and 30 t ha$^{-1}$ sheep manure were distributed on the field and mixed with soil. To accelerate and complete the decay process portion of 100 kg ha$^{-1}$ urea was added to the manures. Then, vermicompost was distributed and mixed with soil. Vermicompost prepared from livestock manure and earthworm species in Zahak (southeastern Iran) from the research farm of Zabol University, Iran, and in Fariman (northeastern Iran) from the Kaveh Support Services Company in Mashhad, Iran. Table S1 presents the chemical properties and elements composition in the vermicompost fertilizer and manure samples.

Field preparation and planting occurred in the second half of February when the soil temperature was sufficient (over 20 °C at both locations). Immediately the field was prepared as the ridge planting. Planting occurred on both sides of the ridges. The bed

width was 3 m, and the distance between the rows was 70 centimeters. Therefore, the area of each plot was 21 square meters. The depth and width of the furrows were 50 and 60 centimeters. After that, a portion of 100 kg ha$^{-1}$ urea, 100 kg ha$^{-1}$ triple superphosphate, and 150 kg ha$^{-1}$ potassium sulfate were distributed and mixed with soil. Then, 3 kg ha$^{-1}$ native melon seeds of the Khatouni variety using for planting.

The first furrow irrigation occurred before seed planting. The seeds germinated using soil moisture and turned green within one week. Then, the second irrigation occurred. The irrigation was done every five days, except under certain conditions, such as high temperatures for several days, which reduced the irrigation interval every three days.

Nanobiomic and silicone were sprayed in 2 and 3 liters (per thousand liters of water) per hectare at the four-leaf stage, respectively. Nanobiomic biofertilizer containing nitrogen-fixing bacteria (*Azotobacter chorococum* and *Azospirillum lipoferoum*) and phosphorus-dissolving bacteria (*Pseudomonas putida* and *Bacillus lentus*) with $10^8$ living cells per gram, 32% humic acid, 2% folic acid, 0.1% molybdenum, 12% potassium, 0.36% magnesium, 4.3% manganese, 0.36% calcium, 10% zinc, 5.9% iron, and various amino acids. Silicon solution, with the formula of silicon oxide, was used as silicic acid ($H_4SiO_4$) with a composition of 30% by weight and 36% by volume.

## Harvesting operation

Fruit harvesting was done in one week following physiological ripening and detecting changes in color or latticing on fruits on June 26 in southeastern Iran (Zahak county) and August 7 in northeastern Iran (Fariman county).

## Modeling methods

### Support vector machine (SVM)

*Boser, Guyon & Vapnik (1992)* presented the support vector machine as a learning tool for both regression and classification. Over the next few years, they offered an optimum primal theory as a linear classifier and used kernel functions to develop non-linear classifiers. Finally, in *1995*, *Vapnik* developed support vector regression (SVR). The SVR derives from statistical training theory for minimizing the risk structure (*Vapnik, 1998*). Data classification issues are solved using the SVM classification model, while prediction problems are solved using the SVR model.

### Support vector regression (SVR)

The accuracy of the performance function is an essential issue in probabilistic modeling approaches-based reliability analysis. The SVR model applying successfully in reliability analysis (*Dai et al., 2012*) using the simulation approaches (*Sun, Wang & Li, 2017*) and partial sampling. Thus, the hybrid and conjugate forms of SVR can provide efficient and accurate results of reliability analysis-based spermophagy. Consequently, the SVR modeling approach to build the nonlinear relation can improve the accuracy in predicting the probabilistic model with random input parameters $X$. The spermophagy (SP) is structured as the Eq. (1):

$$SP = b + \sum_{i=1}^{N} w_i K(x, x_i) \tag{1}$$

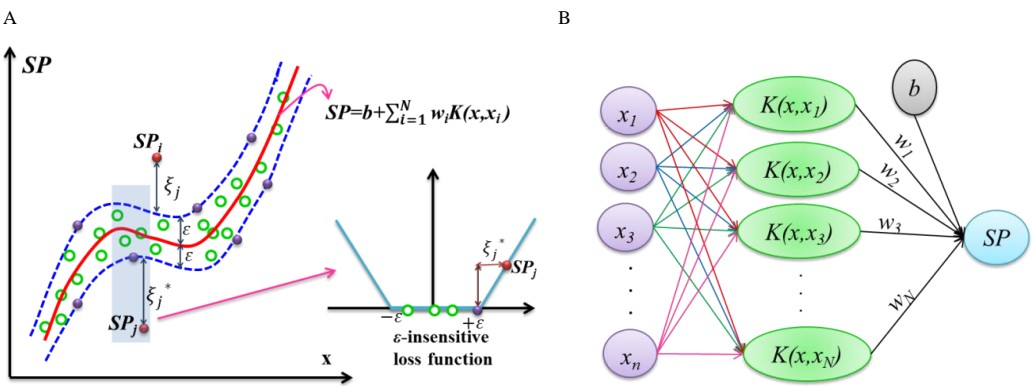

**Figure 1** **Schematic view of probabilistic model-based SVR.** (A) Calibrating data with the $\epsilon$-insensitive loss function. (B) Structure of SVR for predictions of spermophagy.

where $b$ is bias, and $K(x, x_i)$ represents the kernel function for transferring the input variables from real space into $N$-dimensional feature space. Generally, the Gaussian kernel function uses for transferring the input data as follows in Eq. (2) (*Brereton & Lloyd, 2010*):

$$K(x, x_i) = \exp(-0.5\|x - x_i\|^2/\sigma^2) \qquad (2)$$

where $\sigma$ is the kernel parameter that provides the smoothness of the kernel function, given as $\sigma = 0.5$. $w_i$ is the weight to connect the input random data points in feature space and spermophagy (SP) for computing by use of two slack variables $\xi_i, \xi_i^*$ by the following optimization problem in Eq. (3) (*Lu, 2014*):

$$Minimise \; \frac{\|w\|^2}{2} + C \sum_{i=1}^{N} (\xi_i + \xi_i^*)$$
$$Subjected \; to \begin{cases} y_i - <w.K(x, x_i)> - b \leq \varepsilon + \xi_i \\ <w.K(x, x_i)> + b - y_i \leq \varepsilon + \xi_i^* \\ \xi_i, \xi_i^* \geq 0 \end{cases} \qquad (3)$$

In which factor $C \geq 0$ is the regularization coefficient given as $C = 500$ and $\varepsilon$ is the insensitive loss function given as $\varepsilon = 0.01$ in this study. The $\varepsilon$- insensitive loss function uses to neglect the calibrating process-based SVR when differences between the predicted and observed SP is less than $\varepsilon$ schematically shown in Fig. 1A. The structure of SVR represents in Fig. 1B, and the input data set $(x)$, such as fruit yield and nitrogen, phosphorus, and potassium in the seeds, fruits, leaves, and roots, are used to calibrate the probabilistic model of SP using SVR.

### Identification accuracy

The means of standard deviation (SD) and root means squared error (RMSE) as the accuracy measures, mean absolute percentage error (MAPE), the ratio of performance to deviation (RPD), Pearson correlation coefficient (r), coefficient of determination ($R^2$), and coefficient of variation (CV) uses as the goodness of fit measures to prediction models in this study.

*Used software*

Matlab V7.1 software (The Mathworks Inc., Natick, Massachusetts, USA) uses for regression analysis and prediction models of fruit yield and macro-nutrient content in plant organs of *Cucumis melo* in response to soil elements. Also, excel software using for drawing figures of the above-described parameters.

# RESULTS AND DISCUSSION

## Investigating the predictive models of the fruit yield and nitrogen content in plant organs

Based on the results, the statistical parameters of observed values input to the model represents in Table S2, and the predicted values obtained from the model are in Table S3. The results of fitting the predicted values in the fruit yield and seed, fruit, leaf, and root nitrogen of the melon compared to observed values in response to soil nitrogen based on the SVR model represents in Table S4.

According to the estimated parameters, the predicted fruit nitrogen values have the highest accuracy (RMSE = 0.122; MAPE = 7.01) in the model fitting, while the leaf nitrogen values have the lowest accuracy (RMSE = 1.061; MAPE = 31.85). The RPD statistic evaluates the model's performance. Values less than 1.4, between 1.4 and 2, and greater than 2: respectively, show weak, acceptable, and excellent modeling performance (*Chang et al., 2001*). Accordingly, the fruit nitrogen had an excellent modeling performance (RPD = 2.017) in the prediction, and the leaf nitrogen had a weak performance (RPD = 0.710). However, the regression model obtained from leaf nitrogen values ($R^2 = 0.832$; Adjusted $R^2 = 0.831$; Beta = 0.912) was the most suitable prediction model, followed by fruit nitrogen values ($R^2 = 0.807$; Adjusted $R^2 = 0.805$; Beta = 0.898). In contrast, the model obtained from root nitrogen ($R^2 = 0.542$; Adjusted $R^2 = 0.539$; Beta = 0.736) had the weakest performance in prediction. The closer these values are to one, the model indicates a stronger correlation between the predicted values and the observed values. In other words, the regression model obtained from the leaf and fruit nitrogen prediction can cover or express a higher percentage of observed values. Also is known that the coefficients of each variable are positive, and due to the significant value of each variable being smaller than 0.05 (Sig = 0.000 < 0.05), this is proof of the appropriateness of the obtained models. Any variable with the larger Beta has more importance in the regression model. In this way is known that leaf nitrogen (Beta = 0.912) followed by fruit nitrogen (Beta = 0.898) will be the best variables for predicting plant nitrogen changes in response to soil nitrogen (Table S4). *Seidel et al. (2019)* used spectrometry to evaluate organic carbon and nitrogen of whole rangeland soils in Germany; they used a simple regression model to estimate these soil properties and assessed organic carbon and total nitrogen with acceptable accuracy ($R^2 = 0.65$ and RPD = 2.7) and excellent accuracy ($R^2 = 0.87$ and RPD = 2.7), respectively.

Table S4 presents the correlation between the observed and predicted values in fruit yield and plant organs using the SVR method. The results showed the high potential of the SVR algorithm in predicting the observed values in fruit yield and plant organs. The predictive performance of the SVR algorithm for leaf and fruit nitrogen values is better than seed, root nitrogen, and fruit yield values.
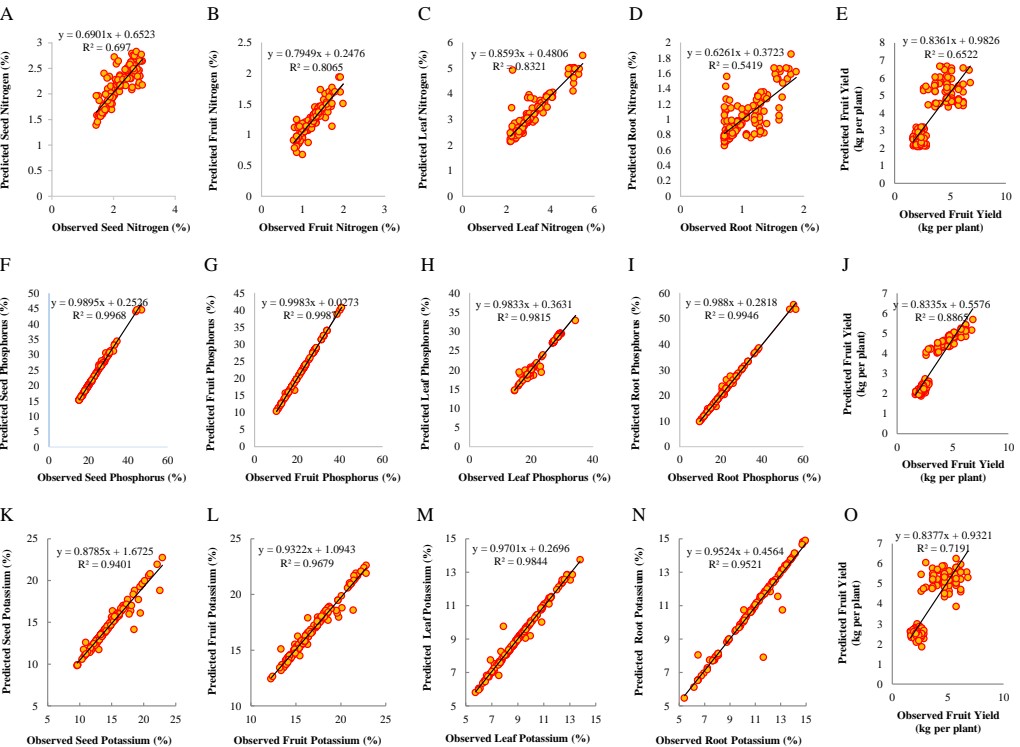

**Figure 2  Scatter diagrams of observed and predicted values of macro-nutrient content in plant organs and fruit yield for the training data ($N = 144$).** Scatter diagrams of observed and predicted values of nitrogen represented in A–D, phosphorus in F–I, and potassium in K–N for seeds, fruits, leaves, and roots in response to soil elements. E, J, and O represent fruit yield in response to soil nitrogen, phosphorus, and potassium.

Scatter diagrams for predicted values of nitrogen content in seed, fruit, leaf, and root, and fruit yield compared to observed values in response to soil nitrogen are presented in Figs. 2A–2E respectively. The regression line slope of diagrams for investigated plant nitrogen values in the SVR model represents in these figures. Predicted nitrogen values in the leaf and fruit had the lowest distance from the 1:1 line and the best fitting based on these results. The seed, root nitrogen, and fruit yield prediction had the highest relative distance from the 1:1 line and the lowest accuracy. The scatter of dots in the figures indicates the models' accuracy in predicting the values of nitrogen output. Consequently, it observed a strong positive correlation and acceptable accuracy between the data and the model (Figs. 2A–2E).

Figures 3A–3E presents the diagrams for plant nitrogen and fruit yield changes in response to soil nitrogen values. In general, by increasing soil nitrogen, the nitrogen content of different organs increases to maximize plant growth. Nevertheless, increasing the amount of soil nitrogen observed a reduction in fruit yield, probably due to the negative impacts of excess nitrogen. Through yield reduction by increasing soil nitrogen, we can also refer to Mitscherlich's law, in which the plant is provided with a certain fertilizer level or growth-limiting element to achieve a good yield. According to the results, the best soil

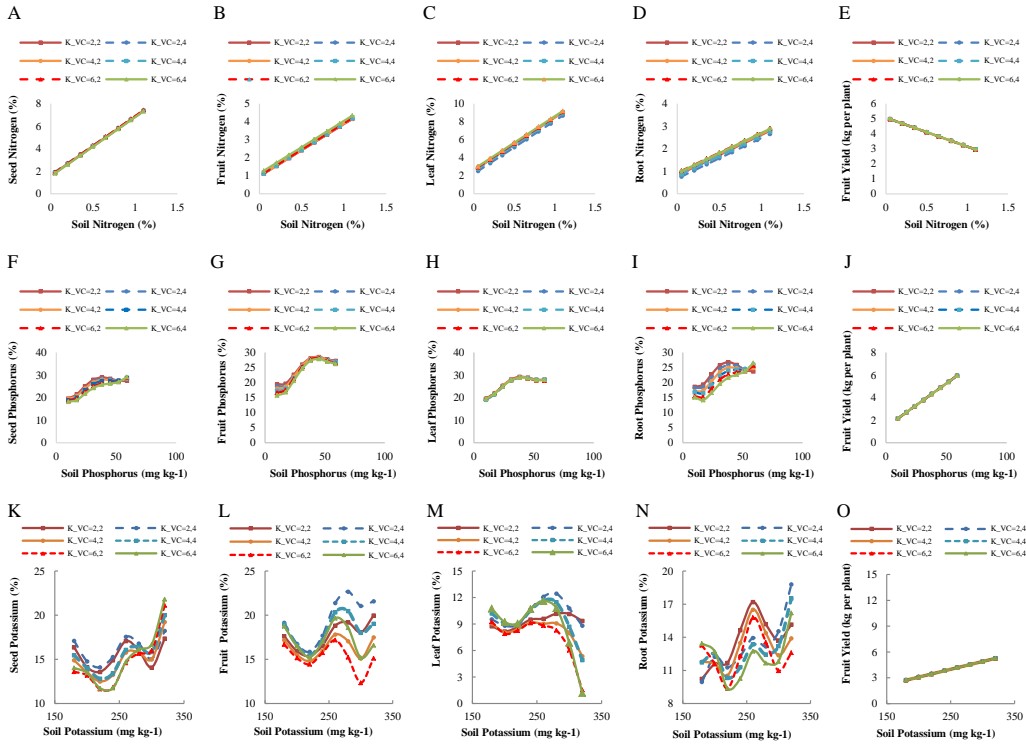

**Figure 3** **Patterns of changes in the predicted values of macro-nutrient in plant organs and fruit yield in response to soil elements under different fertilizer and vermicompost levels.** Patterns of changes in the predicted values of nitrogen represented in A–D, phosphorus in F–I, and potassium in K–N for seeds, fruits, leaves, and roots in response to soil elements. E, J, and O represent fruit yield in response to soil nitrogen, phosphorus, and potassium. The use of cow manure + 5 t ha$^{-1}$ of vermicompost (K_VC = 2,2); cow manure + 15 t ha$^{-1}$ of vermicompost (K_VC = 2,4); Nanobiomic foliar application + 5 t ha$^{-1}$ of vermicompost (K_VC = 4.2); Nanobiomic foliar application + 15 t ha$^{-1}$ of vermicompost (K_VC = 4.4); use of chemical fertilizer + 5 t ha$^{-1}$ of vermicompost (K_VC = 6,2); the use of chemical fertilizer + 15 t ha$^{-1}$ of vermicompost (K_VC = 6,4).

nitrogen ranged between 0.05 and 1.1% to obtain the most accurate predictions of the plant nitrogen content. According to the results of the predictions, the highest increase in the plant nitrogen in response to soil nitrogen content ranged from 3.04 to 9.18% for the leaf nitrogen and from 1.27 to 4.33% for the fruit nitrogen under NPK chemical fertilizers and using 15 t ha$^{-1}$ of vermicompost. Then, changes in the root nitrogen content were predicted in the range of 1.017 to 2.90% under NPK chemical fertilizers by 5 t ha$^{-1}$ of vermicompost. Also, changes in the seed nitrogen content ranged from 1.93 to 7.39% under cow manure and using 15 t ha$^{-1}$ of vermicompost. The changes in fruit yield ranged from 2.95 to 4.97 kg per plant under NPK chemical fertilizers and using 15 t ha$^{-1}$ of vermicompost (Figs. 3A–3E).

According to the performance evaluation of predicted models, the nitrogen content in the leaves and fruits is better than that in the seeds, roots, and fruit yield, so they founded to be more suitable for crop monitoring. The predictions showed that despite the error in soil measurements and the effectiveness of nitrogen values from a combination of different

parameters related to the crop, there is a strong linear correlation between the crop's nitrogen content and the soil nitrogen values. According to the diagrams, there was a slight difference between the predicted and the observed values. Since the crop's nitrogen content is closely related to soil nitrogen values, the observed values are approximately equal to the estimated nitrogen values. The observed incremental relationship between the crop's nitrogen content and soil nitrogen values is calculated by regression equations, as shown in Figs. 3A–3E. The results of this study are consistent with those of *Dotto et al. (2018)*.

Nitrogen is a macro-nutrients for plant growth, so determining its amount and changes in organic compounds is critical for evaluating the final fertilizer's value. Fertilization seems to have increased nitrogen content in the seeds, fruits, leaves, and roots. Due to the similar trend of leaf and fruit nitrogen changes, this result indicates that under NPK chemical fertilizers and using 15 t ha$^{-1}$ of vermicompost, followed decomposition process of organic matter by microorganisms and earthworms. Thus, the nitrogen content of the plant's vegetative body has increased by improving photosynthesis and retransferring of photosynthetic materials; more nitrogenous compounds have been transferred to the fruit and have increased the percentage of fruit nitrogen.

This result is consistent with other research findings. *Tang et al. (2013)* reported that the amount of nitrogen in citrus leaves was significantly related to the amount in the soil. In addition, increasing the yield of bitter cucumber with the application of nitrogen, phosphorus, and potassium fertilizers by *Baset Mia et al. (2014)* has been reported. The findings of other researchers also show the positive effect of vermicompost fertilizer on plant characteristics (*Simon & Bababbo, 2015*).

## Investigating the predictive models of the fruit yield and phosphorus content in plant organs

The statistical parameters of the observed values input to the model represent in Table S5, and the predicted values obtained from the model are in Table S6. The results of fitting the predicted values in the fruit yield and seed, fruit, leaf, and root phosphorus compared to observed values in response to soil phosphorus based on the SVR model represents in Table S7.

The "regression" refers to obtaining a hyperplane that fits the data. The distance of each dot from this hyperplane indicates the error of that particular dot. According to the predicted results, fruit phosphorus had the lowest error (RMSE = 0.228; MAPE = 0.38%); and leaf phosphorus had the highest error values (RMSE = 22.98; MAPE = 90.57%) in the model's fitting. Based on the ratio of performance to deviation results, fruit phosphorous had excellent performance (RPD = 27.95), and leaf phosphorus had weak performance (RPD = 0.208) in model prediction (Table S7).

The regression coefficients obtained from the model's fitting for the seed, fruit, leaf, root phosphorous, and fruit yield were 0.997, 0.999, 0.981, 0.995, and 0.886, respectively, which showed a strong positive correlation between the predicted and observed values. Thereby, fruit phosphorous ($R^2 = 0.999$) had the highest contribution in prediction than observed values, thus identified as the best model compared to other models. Consequently,
the model fitting results indicate the success of the SVR model in predicting the plant phosphorus compared to observed values in response to soil phosphorus (Table S7).

After investigating the SVR models' accuracy and determining the general correlations between the data, the diagrams for the observed and predicted values of fruit yield and plant organs' phosphorus were drawn (Figs. 2F–2J). The results showed reliable modeling of SVR for predicting the fruit yield and phosphorus content in plant organs. The predicting models' performance of the fruit phosphorus was better than the leaf, root, and seed phosphorus and fruit yield. The scatter diagrams for each parameter are represented in Fig. 2B. Depending on the figures, observed and predicted values were scattered close to the 1:1 line. Consequently, it found a positive correlation with high model accuracy.

Figures 3F–3J represents the diagrams for fruit yield and plant organs' phosphorus changes in response to soil phosphorus content. The most suitable soil phosphorus content was estimated between 10 to 59 mg kg$^{-1}$ to achieve the optimum results in predicting the plant phosphorus. According to the diagrams, the release of phosphorus rate from different fertilizers was slow at first. However, gradually, after the decomposition of fertilizers used in the experiment by releasing nutrients and increasing the soil phosphorus content up to 38 mg kg$^{-1}$, the plant organs' phosphorus values increased to their maximum, then slightly decreased and followed at a constant rate. This pattern of changes is almost the same in all fertilizer and vermicompost treatments. Consequently, the soil phosphorus content of up to 38 mg kg$^{-1}$ will be suitable and sufficient to supply the plant's agronomic needs. More soil phosphorus values do not affect increasing fruit yield and phosphorus content in plant organs, and more applications may not be economical.

According to the results, the highest increase in plant phosphorus in response to soil phosphorus content was predicted in the range of 15.74 to 26.19% for fruit phosphorus and 19.44 to 27.97% for leaf phosphorus under NPK chemical fertilizers and using 15 t ha$^{-1}$ of vermicompost. After that, changes in the root phosphorus were predicted in the range of 15.47 to 25.67% under NPK chemical fertilizers and using 5 t ha$^{-1}$ of vermicompost. Also, changes related to the seed phosphorus were predicted in the range of 18.80 to 28.04% under cow manure and using 15 t ha$^{-1}$ of vermicompost. The changes in the fruit yield ranged from 2.16 to 5.95 kg per plant under NPK chemical fertilizers and using 15 t ha$^{-1}$ of vermicompost (Figs. 3F–3J). It found that the amount of soil phosphorus has caused the adjustment and reduction of the error in estimating the fruit yield and plant organs phosphorus.

Chemical fertilizers have increased phosphorus storage by providing it in soil. Also, using 15 t ha$^{-1}$ of vermicompost in the field has increased the phosphorus availability in plants by increasing the organic matter decomposition and phosphorus mineralization and their conversion into plant-usable form. Vermicompost increases phosphorus uptake by increasing phosphorus solubility, activating microorganisms, secreting organic acids, or stimulating phosphatase activity (Busato et al., 2012). Kakraliya et al. (2017), in the study of the nutritional and biological effects of vermicompost on rice, stated that vermicompost increased nitrogen, phosphorus, and potassium availability. Vermicompost can also increase the amount of absorbed phosphorus (Jumadi et al., 2014).

## Investigating the predictive models of the fruit yield and potassium content in plant organs

The statistical parameters of the observed values input to the model represent in Table S8, and the predicted values obtained from the model are in Table S9. The results of fitting the predicted values in the fruit yield and seed, fruit, leaf, and root potassium compared to observed values in response to soil potassium based on the SVR model represents in Table S10.

The regression coefficients obtained from the model's fitting for leaf potassium ($R^2 = 0.984$) and fruit potassium ($R^2 = 0.968$) had the highest and more accurate coefficient of determination, respectively. The $R^2$ in the root, seed potassium, and fruit yield were 0.952, 0.940, and 0.719, respectively. Accordingly, fruit potassium with the highest ratio of prediction performance to deviation (RPD = 5.174) showed better performance than the values in the root potassium (RPD = 4.420), seed potassium (RPD = 3.577), fruit yield (RPD = 1.637), and leaf potassium (RPD = 0.148), respectively. In addition, in the model fitting, the root of the mean square error and the mean absolute percentage error in the fruit potassium (RMSE = 0.465; MAPE = 1.77%) was less than the leaf potassium (RMSE = 12.148; MAPE = 132.11%). Based on these results, the regression model obtained from the fruit potassium compared to the leaf potassium minimized the error coefficients and performed better in estimating the coefficient of determination. It leads to more accurate output models obtained from observed values in the fruit yield and plant organs in response to soil potassium (Table S10).

Figures 2K–2O presents the observed values compared to the predicted values around the 1:1 line using the SVR model. As shown in Figs. 2K–2O, the data around the 1:1 line are well placed. The high coefficient of determination for the regression line between observed and predicted values of potassium content in the leaf, fruit, root, and seed, and fruit yield with coefficients of 0.984, 0.968, 0.952, 0.940, and 0.719 indicates the appropriate efficiency of this model to describe the trend of plant potassium changes in response to soil potassium (Figs. 2K–2O).

Figures 3K–3O represents the diagram for fruit yield and plant organs' potassium changes in response to soil potassium content. The most suitable soil potassium content was estimated between 180 to 320 mg kg$^{-1}$ to achieve the optimum results in predicting the plant potassium. According to the diagrams, at the beginning of growth, due to potassium uptake by the plant, soil potassium decreased and showed a downward trend. However, gradually after the decomposition of fertilizers by releasing nutrients and increasing the soil potassium up to 260–280 mg kg$^{-1}$, the plant organs' potassium values increased to their maximum, then slightly decreased due to the consumption by plant organs. After that, the potassium increased again and reached its maximum in response to 320 mg kg$^{-1}$ of soil potassium. Only the leaf potassium content due to the transfer of nutrients to the fruits and seeds continued to decrease (Figs. 3K–3O).

Because the potassium in the leaf and fruit play an important role in estimating the plant's potassium content, they were identified as the best parameters in the final prediction of plant potassium values in response to soil potassium. According to the prediction results, the highest increase in plant potassium in response to soil potassium ranged from 15.19

to 19.67% for the fruit potassium and 1.18 to 11.60% for the leaf potassium under NPK chemical fertilizer and using 15 t ha$^{-1}$ vermicompost. After that, changes related to the root potassium values ranged from 9.37 to 15.78% under NPK chemical fertilizers and using 5 t ha$^{-1}$ of vermicompost. Also, the changes related to the seed potassium were predicted in the range of 14.09 to 18.22% under cow manure and using 15 t ha$^{-1}$ of vermicompost. The changes in fruit yield ranged from 2.72 to 5.26 kg per plant under NPK chemical fertilizers and using 15 t ha$^{-1}$ of vermicompost (Figs. 3K–3O).

In general NPK chemical fertilizer and using 15 t ha$^{-1}$ vermicompost improved the physical and chemical soil properties and plant nutritional status and increased the absorbable potassium in soil and plants. In this regard, *Fallah, Ghalavand & Raisi (2013)* reported that the use of organic and integrated fertilizers leads to improved vegetative and reproductive parameters due to improving the physical and chemical properties of soil and availability and simultaneous release of essential elements with plant needs, which ultimately enhances the crop yield.

*Cheng et al. (2016)* have stated that available potassium is one of the most important soil factors affecting the yield and quality of Novell orange fruit. In a study by *Xu et al. (2016)* on the response of rice yield to potassium uptake, these researchers attributed the high yield changes to differences in climatic conditions and soil nutrient supply.

## Predictive performance of the SVR model

In this study, an SVR structure using the hybrid and conjugate forms construct to build the nonlinear relation in predicting the probabilistic model trained with input parameters such as fruit yield and nitrogen, phosphorus, and potassium in the seeds, fruits, leaves, and roots. The structure of SVR represents in Fig. 1B. In this way, 75% of the samples ($n = 108$) uses to obtain samples concerning experimental design. The remaining 25% of the samples ($n = 36$), including cow manure + 5 t ha$^{-1}$ of vermicompost (K_VC = 2,2), cow manure + 15 t ha$^{-1}$ of vermicompost (K_VC = 2,4), nanobiomic foliar application + 5 t ha$^{-1}$ of vermicompost (K_VC = 4.2), nanobiomic foliar application + 15 t ha$^{-1}$ of vermicompost (K_VC = 4.4), chemical fertilizer + 5 t ha$^{-1}$ of vermicompost (K_VC = 6,2), and chemical fertilizer + 15 t ha$^{-1}$ of vermicompost (K_VC = 6,4) uses to verify the model. The input variables in the model were repeated three times and normalized to prevent the negative impact of different ranges of input variables on the model's efficiency.

The root means squared error (RMSE) and the coefficient of determination ($R^2$) uses to estimate the training and testing performance, to evaluate the prediction models of fruit yield and macro-nutrient content in plant organs using SVR. A scatter diagram for each parameter constructs between the predicted and observed values, which are scattered close to the 1:1 line (Figs. 2A–2E, 2F–2J and 2K–2O). The $R^2$ between the predicted observed values was 0.807 for fruit nitrogen, 0.999 for fruit phosphorus, and 0.968 for fruit potassium. Also, the RMSE (0.122 for fruit nitrogen, 0.228 for fruit phosphorus, and 0.465 for fruit potassium) was less than the other parameters (Tables S4, S7, and S10, respectively). The SVR model showed fast training and reliable simulation accuracy in model testing. The relationship between the predicted and observed values of fruit yield

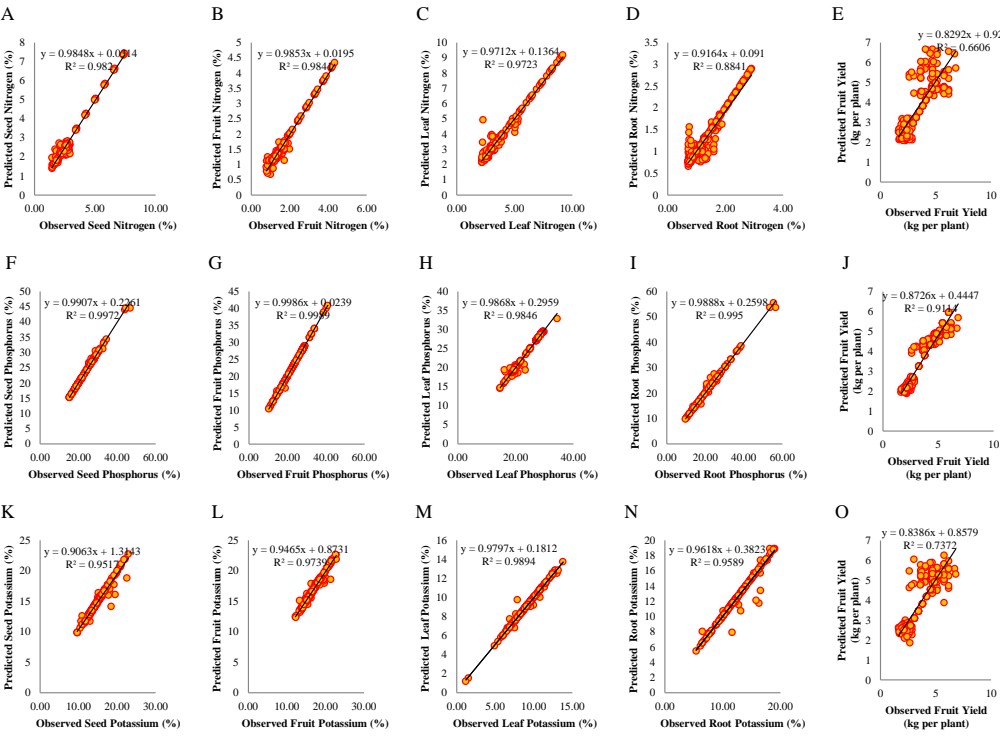

**Figure 4 Scatter diagrams of observed and predicted values of macro-nutrient content in plant organs and fruit yield for the test data ($N = 192$).** Scatter diagrams of observed and predicted values of nitrogen represented in A–D, phosphorus in F–I, and potassium in K–N for seeds, fruits, leaves, and roots in response to soil elements. E, J, and O represent fruit yield in response to soil nitrogen, phosphorus, and potassium.

and macro-nutrient content in plant organs was favorable, indicating that this model has a high degree and consistency of simulation.

To evaluate the predictive performance from the empirical equations of the SVR-based was used for the prediction of fruit yield and macro-nutrient content in plant organs through Eqs. (1), (2) and (3), mentioned in the modeling methods section. To further evaluate the model's generalizability, the developed SVR model was tested using 192 new datasets. The root means squared error (RMSE) and the coefficient of determination ($R^2$) uses to estimate. A scatter diagram for each parameter constructs between the predicted and observed values, which are scattered close to the 1:1 line (Figs. 4A–4E, 4F–4J, and 4K–4O). The $R^2$ between the predicted observed values was 0.984 for fruit nitrogen, 0.999 for fruit phosphorus, and 0.974 for fruit potassium. Also, the RMSE (0.105 for fruit nitrogen, 0.197 for fruit phosphorus, and 0.402 for fruit potassium) was less than the other parameters (Tables S11, S12, and S13, respectively). Therefore, the predictive performance of the SVR model was relatively high and showed a nonlinear fitting ability for predicting fruit yield and macro-nutrient content in plant organs of *Cucumis melo*.

## CONCLUSIONS

This study investigates the prediction models of fruit yield and macro-nutrient content in plant organs of *Cucumis melo* in response to soil elements affected by different fertilizers using support vector regression (SVR). Support vector regression can effectively calibrate input data sets such as fruit yield and nitrogen, phosphorus, and potassium in the seeds, fruits, leaves, and roots to model (Fig. 1). The results showed reliable modeling for support vector regression in predicting the fruit yield and macro-nutrient content in plant organs.

According to the results, when the data sets of the nitrogen, phosphorus, and potassium in the fruit uses as input, the accuracy of these models was higher than 80.0% ($R^2 = 0.807$ for fruit nitrogen; $R^2 = 0.999$ for fruit phosphorus; $R^2 = 0.968$ for fruit potassium) (Tables S4, S7, and S10, respectively). Likewise, the ratio of prediction performance to deviation (RPD) obtained from the models ranged from 2.017 for predicting fruit nitrogen (Table S4), 5.17 for fruit potassium (Table S10), to 27.95 for fruit phosphorus (Table S7) content. Because the macro-nutrient content in the fruit had the highest contribution in prediction than observed values, thus identified as the best model compared to other models in response to soil elements. Based on our findings, the importance of fruit phosphorus identifies as a determinant that strongly influenced melon prediction models.

According to the results of the prediction models in response to soil elements, the best soil nitrogen content ranged from 0.05 to 1.1%, soil phosphorus from 10 to 59 mg kg$^{-1}$, and soil potassium from 180 to 320 mg kg$^{-1}$, which offers a suitable macro-nutrient content in the prediction models. Likewise, the best fruit nitrogen content ranged from 1.27 to 4.33%, fruit phosphorus from 15.74 to 26.19%, fruit potassium from 15.19 to 19.67%, and fruit yield from 2.16 to 5.95 kg per plant obtained under NPK chemical fertilizers and using 15 t ha$^{-1}$ of vermicompost. More significant values of soil elements do not affect increasing fruit yield and macro-nutrient content in plant organs, and excessive application may not be economical. Therefore, predicting the macro-nutrient content in the fruits of *Cucumis melo* in response to soil elements caused a saving in the amount of fertilizer utilized and provides for the possibility of proper farming activities during the growing season.

## ACKNOWLEDGEMENTS

We are grateful to the field technicians at the research farm of Baqiyat-Allah-ul-Azam Agricultural Research Institute of Zabol University, Zahak, Iran, and the faculty members of Zabol University, Zabol, Iran, as well as the field technicians at Fariman, Iran, and the faculty members of Ferdowsi University, Mashhad, Iran, who helped in the fieldwork.

### Funding
The authors received no funding for this work.

### Competing Interests
The authors declare there are no competing interests.

## Author Contributions

- Abbas Keshtehgar conceived and designed the experiments, performed the experiments, analyzed the data, prepared figures and/or tables, authored or reviewed drafts of the article, and approved the final draft.
- Mahdi Dahmardeh conceived and designed the experiments, performed the experiments, analyzed the data, prepared figures and/or tables, authored or reviewed drafts of the article, and approved the final draft.
- Ahmad Ghanbari conceived and designed the experiments, performed the experiments, authored or reviewed drafts of the article, and approved the final draft.
- Issa Khammari conceived and designed the experiments, analyzed the data, prepared figures and/or tables, and approved the final draft.

## Data Availability

The raw measurements are available in the Supplementary Files.

## Supplemental Information

Supplemental information for this article can be found online at http://dx.doi.org/10.7717/peerj.15417#supplemental-information.

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
