# Peer review of "Prediction models of macro-nutrient content in plant organs of Cucumis melo in response to soil elements using support vector regression"

_PeerJ, doi:10.7717/peerj.15417_

## Round 0.1 · original submission · Major Revisions

Dear Authors,

Three independent experts have evaluated your work. They all agreed that the paper could be published in PeerJ, but significant revisions had to be made beforehand. Please read the comments of the reviewers and respond to them.

With best regards,

·

Basic reporting

Introduction part should be concise and little brief. The finding of earlier work done presented in introduction from 75 to 135 may be discussed in discussion part rather than in introduction.

Experimental design

Sufficient

Validity of the findings

Conclusions are well stated, linked to original research question & limited to supporting result

Additional comments

Please see attachment

·

Basic reporting

• The manuscript support interesting and important results about support vector regressions involving soil macro-nutrient concentrations, and seed, root, leave and fruit macro-nutrient contents. These regressions take as response (predicted) each of the mentioned plant’s variables. Nonetheless, yield as final variable was not considered. I suggest that, if it is possible, authors must involve yield per plant as predicted variable.
• The present manuscript must be written in correct English at professional standard level. I have inserted some questions and possible changes to improve the grammar English.
• Figures and Tables are relevant to the content of the article. However, some of them must be designed according to the Peer Journal author guidelines, and correct some words (for example, the word Phosphore must be corrected in the Figures 4 and 5).

Experimental design

• The manuscript describes original primary research within the Aims and Scope of the Peer Journal.
• The manuscript clearly defines the research focus, and it is relevant and meaningful.
• Methods information should be described more clearly, mainly those referred to statistical estimators.
Accuracy is the degree of how close a calculated or measured value is to the actual value. It measures the statistical error, that is, the difference between the measured value and the actual value. The range in those values indicates the accuracy of the measurement. It is common to measure accuracy by determining the average value of multiple measurements. When working with a set of data, it is also important to calculate the precision of those measurements to ensure accurate results (https://www.indeed.com/career-advice/career-development/how-to-measure-accuracy).
The accuracy formula provides accuracy as a difference of error rate from 100%. To find accuracy we first need to calculate the error rate. And the error rate is the percentage value of the difference of the observed and the actual value, divided by the actual value.
Accuracy = 100% - Error Rate
Error Rate = |Observed Value - Actual Value|/Actual Value × 100
Precision measures how close the various measurements are to each other. You can measure precision by finding the average deviation, that is, the average of the differences in measurements. Data can be precise without being accurate, but ideally, the measurements are both precise and accurate to produce quality results.
Moreover, the word “precision” does not have a single meaning. You can represent precision using several different measurements.
I. Range. For small data sets with about ten or fewer measurements, the range of values is a good measure of precision. This is particularly true if the values appear reasonably closely grouped. If you see one or two values that appear far from the others, you may wish to use a different calculation.
II. Average deviation. The average deviation is a more accurate measure of precision for a small set of data values.
III. Standard deviation. The standard deviation is perhaps the most recognized measure of precision. Standard deviation may be used to calculate the precision of measurements for an entire population or a sample of the population.
The Root Mean Squared Error (RMSE) is one of the two main performance indicators for a regression model. It measures the average difference between values predicted by a model and the actual values. It provides an estimation of how well the model is able to predict the target value (accuracy).
Markedly, some statistical estimators are used as accuracy measures and other as goodness of fit criteria. For instance, RMSE is an absolute measure of fit and R-squared is a relative measure of fit or goodness of fit criterion. Then, the authors must clearly point out which statistical estimators they used as accuracy measures and as goodness of fit measures in the Materials and Methods section.
• The evidences appreciated in the manuscript suggest that research work has been conducted in conformity with the prevailing ethical standards in the field.

Validity of the findings

• Evidences indicate that the data were well developed and used.
• The conclusions are connected to the original question investigated, and limited to those supported by the results.

Additional comments

Your results are interesting and important; so, I suggest you should improve the english version of your manuscript and include information on questions inserted in the manuscript file, and invlve yield as predicted variable.

·

Basic reporting

In the material-method, the methods related to the parameters presented in Table-1 should be specified.

Experimental design

In material method section, the sizes of the parcels or blocks that make up the experiment should be given.

Validity of the findings

No comment

Additional comments

A successful article in terms of novelty and literature overlap.
However, it is a little long. Some sections should be shortened.

---

## Round 0.2 · Major Revisions

Dear Authors,
Although the reviewers no longer have comments on your work, the Section Editor still has some concerns, which are listed below:

1: There is no indication that the data was split into "training" and "test" sets before modeling. For these types of predictive models it is mandatory that a random set of data be held aside as a "test" set and that this data is not used for training the model. The analysis needs to be repeated with test and training splits and the predictive ability of the model on the test set needs to be reported.

2: The manuscript needs significant editing for grammar and clarity. I have highlighted issues in the abstract in a PDF that I am sending to PeerJ Staff [NOTE: This will be uploaded as a review-only file and forwarded by email] but the whole manuscript needs work.

3: It is unclear from the abstract exactly what the modeling goal is. Predicting nutrient content in plant organs from soil content? I assume so based on the title but this also should be said in the abstract. Is yield also being predicted? From which predictors?

4: Agricultural modeling is described as being a new technique, but agricultural modeling has been going on for decades.

We kindly ask you to respond to these comments and make appropriate corrections to the work.

With best regards,

·

Basic reporting

Clear and unambiguous, professional English used throughout the manuscript. Literature references, sufficient field background/context provided. Professional article structure, figures, tables. Raw data shared properly.

Experimental design

Original primary research within Aims and Scope of the journal. Methods described with sufficient detail & information to replicate.

Validity of the findings

Well defined

Additional comments

Authors have addressed all the comments nicely.

·

Basic reporting

Adequate

Experimental design

Adequate

Validity of the findings

Acceptable.
No comment.

Additional comments

The work is original and generally well discussed. It is suitable for publication in your journal.

---

## Round 0.3 · accepted · Accept

Dear Authors,

Thank you very much for submitting a revised version of the article. In my opinion, all the editor's comments have been taken into account and the work can be published in PeerJ. Congratulations!